# Evolution-aware VAriance (EVA) Coreset Selection for Medical Image Classification

## ABSTRACT

In the medical field, managing high-dimensional massive medical imaging data and performing reliable medical analysis from it is a critical challenge, especially in resource-limited environments such as remote medical facilities and mobile devices. This necessitates effective dataset compression techniques to reduce storage, transmission, and computational cost. However, existing coreset selection methods are primarily designed for natural image datasets, and exhibit doubtful effectiveness when applied to medical image datasets due to challenges such as intra-class variation and inter-class similarity. In this paper, we propose a novel coreset selection strategy termed as *Evolution-aware VAriance (EVA)*, which captures the evolutionary process of model training through a dual-window approach and reflects the fluctuation of sample importance more precisely through variance measurement. Extensive experiments on medical image datasets demonstrate the effectiveness of our strategy over previous SOTA methods, especially at high compression rates. EVA achieves 98.27% accuracy with only 10% training data, compared to 97.20% for the full training set. None of the compared baseline methods can exceed Random at 5% selection rate, while EVA outperforms Random by 5.61%, showcasing its potential for efficient medical image analysis.

## KEYWORDS

Coreset Selection, Medical Image Classification, Evolution-aware Variance

## 1 INTRODUCTION

In the medical field, data collection and processing are essential for delivering accurate and reliable diagnoses and treatment plans. Medical imaging data, typically characterized by high dimensionality and large volumes, necessitates substantial resources for storage and transmission. Moreover, training deep learning models on large-scale medical image datasets requires extensive computational resources and time. This presents challenges in resource-limited settings, such as remote medical facilities where effective medical image analysis is crucial, or on mobile devices where real-time monitoring and analysis are needed. Therefore, efficient data compression and processing techniques become imperative. In this

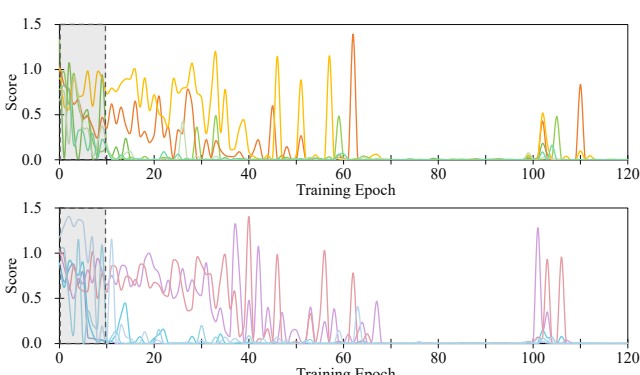

**Figure 1: Existing single-timeframe/window snapshots methods fail to capture the fluctuations of sample importance across epochs. Different samples are denoted in different colors. Here, we measure importance score using the error vector score, a snapshot-based criterion defined in [39], which considers only the first 10 epochs as indicated by the dashed box. These scores are obtained by training ResNet-18 on dataset OrganAMNIST.**

context, coreset selection, or dataset pruning, emerges as a promising approach to mitigate these challenges. Coreset selection condenses a given large-scale dataset into a significantly smaller subset, known as the coreset. The coreset is expected to preserving the essential knowledge of the original full dataset such that the former yields a similar performance as the latter.

Numerous coreset selection works [19, 31, 33, 37, 40, 51, 56] have explored various criteria for identifying important data samples, including geometry distance [44, 50], uncertainty [11], loss [39, 48], decision boundary [13, 32], and gradient matching [35]. However, most of these methods have been validated mainly on natural image datasets, such as CIFAR-10, CIFAR-100 [28], and not extensively on medical datasets. The applicability of those methods for medical image datasets are under exploration, given the unique characteristics of medical images. Compared to natural image datasets, the intra-class variation and inter-class similarity of medical image datasets [46] pose specific challenges to coreset selection. On the one hand, in medical imaging, samples within the same category can exhibit significant differences, making it difficult to capture consistent features for each class. This variation largely comes from the diversity in disease manifestation across patients and discrepancies in imaging conditions. On the other hand, the challenge of inter-class similarity arises when images representing different diseases exhibit similar visual characteristics. Fig. 6 provides a more straightforward demonstration of this characteristic. These factors contribute to the complexity of medical image analysis and underscore the need for sophisticated coreset selection methods that can effectively address these challenges.

To enable the coreset to effectively approximate the model's performance on the full training set with fewer samples, it is essential to consider the training process on the original dataset. This necessitates that the selection methods should effectively capture the varying importance of samples at different training stages. Yosinski et al. [58] highlighted that shallow layers of the network learn general features, while deeper layers learn task-specific features. Han et al. [17] observed that deep models tend to memorize easy instances initially and adapt to harder instances as training progresses. These studies confirm the evolutionary nature of deep learning from simpler to more complex stages. Given these observations[17, 58], we posit that in the domain of medical imaging, the training process of deep learning models exhibits similar characteristics. For instance, in kidney images, the model initially learns the general kidney shape and gradually distinguishes more detailed features of different kidneys. Moreover, as shown in Fig. 1, the significance of samples in enhancing the model performance varies across different training stages [7, 20, 48, 60]. Specifically, certain samples may be crucial for the model's initial learning phase, while others gain importance in the later stages of training.

Most of the existing methods evaluate sample importance using a snapshot of training progress. For example, Xia et al. [51] calculate the distribution distances of features at the end of training. Zhang et al. [60] have proved that the importance scores of samples varies with epochs during training, resulting in significant variations in the constructed coresets at different snapshots. Therefore, methods reliant on single-timeframe snapshots might be inadequate for capturing the comprehensive evolution of model training, overlooking the dynamic characteristics of learning process.

Expanding the scope of the considered training dynamics is a straightforward approach to address this limitation. Previous studies have attempted to incorporate training dynamics using various methods. For example, Pleiss et al. [41] measures the probability gap between the target class and the second-largest class in each epoch; Paul et al. [39] utilize the expected value of error vector scores generated by a few epochs in early training (the first 10 epochs). While this approach partially expands the scope of the considered training dynamics, it overlooks the potential effectiveness of later stages of training, and more importantly, it focuses on samples with high expected error values, indicating that these samples are consistently predicted incorrectly over many training iterations. Such samples may just be too difficult/noisy and may degrade the model performance [7]. Toneva et al. [48] count the number of forgetting events during training, which occur when samples, previously classified correctly, are subsequently predicted incorrectly. However, this counting approach only provides the discrete probability of an event, lacking the granularity needed to reflect the variations of sample contributions throughout the training process.

To address these limitations, in this paper, we propose a novel sample importance scoring strategy called **Evolution-aware VAriance (EVA)**, aiming at achieving reasonable and effective compression of medical image datasets. Firstly, to mitigate the biases from focusing solely on a snapshot or single segment of the training process, we introduce a dual-window approach that considers training dynamics at different stages. This strategy provides a more holistic understanding of the model's learning evolution, enabling

nuanced assessment of sample importance as the model evolves from learning general to specific features. Secondly, within each window, to reflect the fluctuation of sample importance during the model training process in a more precise way, we propose to measure the variance of samples' error vector. The combination of these two strategies provides a more refined and accurate evaluation of sample importance, enabling a more effective coreset selection that aligns with the dynamic nature of neural network training. This approach is particularly beneficial in high compression scenarios for medical image datasets, where maintaining accuracy and reliability is challenging but crucial.

In a nutshell, our contributions can be summarized as follows.

- We identify the limitations of existing coreset selection methods in capturing the evolutionary nature of model training and the fluctuations in sample importance within medical image datasets.

- We thereby propose a novel coreset selection strategy called **Evolution-aware VAriance (EVA)**, which features two key components. The first is a dual-window approach that captures the training dynamics by considering distinct stages of the learning process. The second is the employment of variance measurement on samples' error vectors, offering a granular and more precise evaluation of each sample's contribution to the model training.

- Extensive evaluations on the OrganAMNIST and OrganSMNIST datasets demonstrate that our EVA strategy outperforms SOTA methods at challenging low selection rates while achieving comparable accuracy at high selection rates, showcasing its potential for efficient medical image analysis.

## 2 PRELIMINARIES

In this paper, vectors and matrices are denoted by bold-faced letters. Given a large-scale dataset, we denote the full training set contains N samples as $\mathbb{T} = \{(x_n, y_n)\}_{n=1}^N$, where $x_n \in \mathbb{R}^D$ represents the input feature vector and the corresponding ground-truth label is $y_n \in \mathbb{R}^{1 \times C}$, $C$ is the number of classes. All samples are drawn i.i.d. from a underlying distribution $\mathcal{P}$. We define the neural network as $f_\theta$, parameterized by the weight vector $\theta$. The model output $f_\theta(x_n) \in \mathbb{R}^{1 \times C}$ represents the predicted probabilities of each class. Coreset selection aims to construct a subset (or coreset) $\mathbb{S} = \{(x_m, y_m)\}_{m=1}^M$ ($\mathbb{S} \subset \mathbb{T}$) that captures the essential characteristics of the full dataset, enabling model $f_{\theta^\mathbb{S}}$ trained on $\mathbb{S}$ to achieve comparable or even superior performance compared to model $f_{\theta^\mathbb{T}}$ trained on the entire training set $\mathbb{T}$. The data selection rate $\alpha$ in constructing the coreset is then $\frac{M}{N}$. Under these definitions, following previous work [44], we formulate the objective of coreset selection as,

$$\mathbb{E}_{\substack{(x,y)\sim\mathcal{P}\\ \theta_0\sim\mathcal{G}}} \left[ \ell(x, y; f_{\theta_0}^\mathbb{S}) \right] \simeq \mathbb{E}_{\substack{(x,y)\sim\mathcal{P}\\ \theta_0\sim\mathcal{G}}} \left[ \ell(x, y; f_{\theta_0}^\mathbb{T}) \right] \tag{1}$$

where $f_{\theta_0}^\mathbb{S}$ and $f_{\theta_0}^\mathbb{T}$ represent the neural networks trained on $\mathbb{S}$ and $\mathbb{T}$ with weight $\theta_0$ initialized from distribution $\mathcal{G}$.

## 3 METHODOLOGY

To construct a coreset that satisfies Eq. 1, the error/loss-based approaches propose to measure the contribution of each sample by considering factors such as the loss, gradient, or its influence on

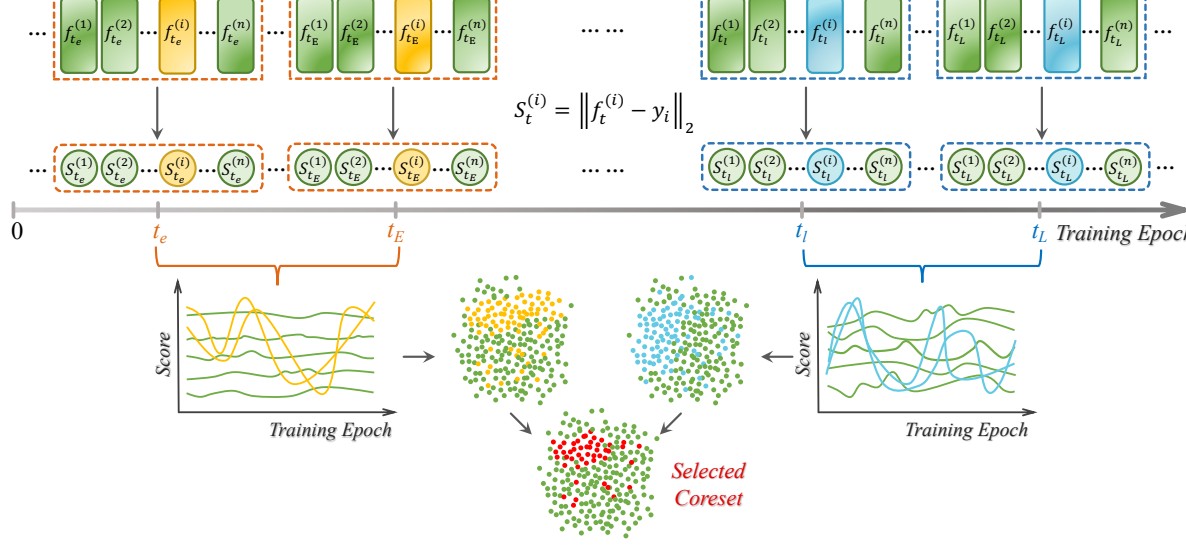

**Figure 2: The pipeline of our proposed EVA . First, we record individual predicted probabilities $f_t^{(i)} = f_{\theta_t}(x_i)$ of samples during training. Then, we measure a score $\mathcal{S}_t^{(i)}$ for each sample, i.e. the L2 norm of error vector. Next, the variance of scores within a window of epochs are calculated to reflect the fluctuation of each sample's contribution. Samples that fluctuate the most are considered important in this stage. Finally, we identify samples that exhibit high importance in dual-window.**

other samples' prediction during model training [16]. In this context, samples that contribute more to the error or loss are considered more important and are thus selected as part of the coreset.

In this section, we delve into the specifics of our proposed Evolution-aware VAriance (EVA) strategy, which comprises two key components. Firstly, we describe how EVA reflects the epoch-level fluctuation by calculating the variance of error-based scores in Sec. 3.1. Following that, we elaborate on how EVA captures the training evolution through a dual-window approach in Sec. 3.2.

## 3.1 Reflecting Epoch-Level Fluctuation via Variance

To approximate the individual contribution of each sample to the reduction in model loss, we initially calculate the variance of error-based scores over a segment of epochs. This process can be further divided into the following steps.

**Step 1. Single Epoch Scoring.** In this step, we concentrate on calculating the error score for each sample at a specific epoch across multiple independent runs. Specifically, for each sample $(x_i, y_i)$ in the training set, we first consider a single epoch $t$ and compute the total mean square error (MSE) across all categories using the equation below:

$$\text{MSE}_t^{(i)} = \sum_{j=1}^{C} (\hat{y}_{ij} - y_{ij})^2, \tag{2}$$

where $\hat{y}_i = f_{\theta}(x_i)$, therefore $\hat{y}_{ij}$ denotes the model output of the $i$-th sample for the $j$-th category, and $y_{ij}$ is the one-hot encoding of the ground-truth label for the $i$-th sample in the $j$-th category. Then, we take the square root of the total MSE for each sample.

Thus, for each sample $(x_i, y_i)$ at epoch $t$, we have the L2 norm of the error vector, representing the discrepancy between model predictions and ground-truth labels:

$$\mathcal{S}_t^{(i)} = \|f_{\theta}(x_i) - y_i\|_2, \tag{3}$$

This process yields a sequence of error scores, providing insights into the prediction performance of the model across different training iterations.

**Step 2. Variance Across Multiple Epochs.** Having obtained the error scores for each sample at individual epochs, in this step, we proceed to assess the variability of scores across multiple epochs by calculating the variance over a segment of epochs. Specifically, for each sample $(x_i, y_i)$, we analyze the training dynamics over a span of $K$ epochs, from $t$ to $t + K - 1$. The error-based scores for this period are represented as $\left\{ \mathcal{S}_t^{(i)}, \mathcal{S}_{t+1}^{(i)}, ..., \mathcal{S}_{t+K-1}^{(i)} \right\}$. We then compute the variance of these scores within the $K$-epoch window using the following equation:

$$\mathcal{V}_t^{(i)} = \frac{1}{K} \sum_{k=t}^{t+K-1} \left( \mathcal{S}_k^{(i)} - \mathcal{E}_t^{(i)} \right)^2, \tag{4}$$

where $\mathcal{E}_t^{(i)} = \frac{1}{K} \sum \mathcal{S}_k^{(i)}$ denotes the mean value within the $K$-epoch window. This calculation provides insight into the consistency or variability of the error-based scores for each sample over a specified segment of training epochs, enabling a more precise understanding of the subtle fluctuations in a sample's impact on model performance over time.

## 3.2 Capturing Training Evolution with Dual-Window

As mentioned in Sec. 1, snapshot-based methodologies often fall short in capturing the comprehensive evolution of model training, thus warranting an expansion in the scope of considered training dynamics. One approach to broaden the scope is to sample some epochs during the training dynamics. However, random or probabilistic sampling of epochs may not effectively capture the dynamic changes in sample importance throughout the entire training process. Another method is to consider epochs within a certain window, as we did in Eq. 4. Nevertheless, this approach carries the risk of excessive bias towards specific training phases.

Therefore, we introduce a dual-window approach to capture the evolution of the training process more comprehensively. The first window focuses on the early stages of training, during which the model primarily learns general features. Samples that significantly impact the overall model performance are likely to exhibit high importance during this stage. The second window targets the later stages of training, where the model gradually learns more specific task-related features. The importance of samples that have a significant impact on the overall model performance may increase or decrease during this stage. By integrating information from dual windows, we aim to identify samples that exhibit high importance in both early and late stages. This implies that these samples contain both general features and specific task-related features. Additionally, the continuous sequence of epochs provides more temporal information, allowing for a more comprehensive assessment of sample importance throughout the entire training process. Overall, the use of two windows provides a more nuanced understanding of training dynamics and sample importance, enhancing the effectiveness of the selection process for constructing a coreset. This effectiveness has been proved in Sec. 4.3.

To maintain consistency with Sec. 3.1, in the dual-window scenario, we also consider windows spanning $K$ epochs. We define the total number of training epochs as $T$, the first window ranges from $t_e$ to $t_E = t_e + K - 1$, and the second window ranges from $t_l$ to $t_L = t_l + K - 1$. These windows are non-overlapping ($t_E < t_l$). Specifically, for each sample $(x_i, y_i)$, we compute the scores within each window of epochs, denoted as $\left\{ \mathcal{S}_k^{(i)} \right\}_{k=t_e}^{t_E}$ and $\left\{ \mathcal{S}_k^{(i)} \right\}_{k=t_l}^{t_L}$. The variance of these scores in Eq. 4 can be formulated as:

$$\mathcal{V}_e^{(i)} = \frac{1}{K} \sum_{k=t_e}^{t_E} \left( \mathcal{S}_k^{(i)} - \mathcal{E}_e^{(i)} \right)^2,$$
$$\mathcal{V}_l^{(i)} = \frac{1}{K} \sum_{k=t_l}^{t_L} \left( \mathcal{S}_k^{(i)} - \mathcal{E}_l^{(i)} \right)^2, \quad (5)$$

Here, $\mathcal{E}_e^{(i)}$ and $\mathcal{E}_l^{(i)}$ denote the average score of sample $(x_i, y_i)$ in two windows, respectively.

Finally, we aggregate the variances from both windows to identify samples that demonstrate high importance in two stages. Thus the EVA score of each sample can be represented as:

$$\mathcal{V}^{(i)} = \mathcal{V}_e^{(i)} + \mathcal{V}_l^{(i)} \quad (6)$$

We then sort samples in the full training set $\mathbb{T}$ by their EVA score $\mathcal{V}^{(i)}$. Samples with higher scores are deemed more effective

at reducing training loss. Given a selection rate $\alpha$, we select the top-ranked M samples to form the coreset, where $M = \lceil \alpha N \rceil$.

Algorithm 1 provides a detailed explanation of the procedure for the EVA scoring strategy.

---

**Algorithm 1** Evolution-aware VAriance (EVA) Scoring Strategy

---

**Inputs:** Full training set $\mathbb{T} = \{(x_n, y_n)\}_{n=1}^N$; Selection rate $\alpha$; Network $f_\theta$ with weight $\theta$; Epochs $T$; Iteration $I$ pre epoch; Early window $(t_e, t_E)$; Late window $(t_l, t_L)$.

1: **for** $t = 1$ to $T$ **do**
2:     **for** $i = 1$ to $I$, sample a mini-batch $\mathbb{B}_i \subset \mathbb{T}$ **do**
3:        Obtain predicted probabilities $f_{\theta_t}(x_n), x_n \in \mathbb{B}_i$
4:        Calculate $\mathcal{S}_i^{(n)}$ by defined Eq. 3 for each $x_n$
5:        Update $\mathcal{S}_t^{(n)} += \mathcal{S}_i^{(n)}$
6:     **end for**
7:     **if** $t_e \leq t < t_E$ **then**
8:        Calculate $\mathcal{V}_e^{(n)}$ by defined Eq. 5 of early window, $x_n \in \mathbb{T}$
9:     **else if** $t_l \leq t < t_L$ **then**
10:       Calculate $\mathcal{V}_l^{(n)}$ by defined Eq. 5 of late window, $x_n \in \mathbb{T}$
11:     **else if** $t = t_L$ **then**
12:       Update $\mathcal{V}^{(n)}$ by defined Eq. 6 as the EVA score of $x_n$
13:     **end if**
14: **end for**
15: Sort samples by $\mathcal{V}^{(n)}$ in descending order, $x_n \in \mathbb{T}$

**Output:** Top-M samples as the coreset $\mathbb{S} = \{(x_m, y_m)\}_{m=1}^M$

---

## 4 EXPERIMENTS

In this section, we provide a comprehensive set of experiments and analyses to showcase the effectiveness of our proposed Evolution-aware VAriance scoring strategy in diverse scenarios. We start by empirically evaluating the performance of our EVA method by comparing it with other baselines (Sec. 4.2). Subsequently, we conduct a series of ablation studies to investigate the effectiveness of the proposed two main components: variance measurement and dual-window strategy (Sec. 4.3). Additionally, we perform cross-architecture experiments to evaluate the robustness of our coresets, assessing their performance when selected on one architecture and tested on others.

### 4.1 Experiment Setup

**Datasets.** MedMNIST is a large-scale collection of medical images comprising 10 datasets, covering multi-modal, diverse data scales (from 100 to 100,000) and classification tasks. The classification performance of this public large-scale datasets has been validated as effective in [54]. More details about MedMNIST are included in Sec. 5.1. In this work, considering the time-consuming training, the effectiveness of the proposed method is primarily evaluated on two 2D datasets from MedMNIST: OrganAMNIST and OrganSM-NIST [3, 52], both derived from 3D computed tomography (CT) images from the Liver Tumor Segmentation Benchmark (LiTS). These datasets are designed for multi-class classification tasks, involving 11 body organs with labels including the bladder, femur

**Table 1: Performances of ResNet-18 using various coreset selection methods on MedMNIST medical datasets. All training is repeated 3 times with different random seeds to calculate mean accuracy with standard deviation. The first and second best results in each column are marked in red and blue, respectively.**

| $\alpha$ | OrganAMNIST | | | | | OrganSMNIST | | | | |
|---|---|---|---|---|---|---|---|---|---|---|
| | 2% | 5% | 10% | 20% | 30% | 2% | 5% | 10% | 20% | 30% |
| Full dataset | 98.39 $\pm0.02$ | | | | | 91.76 $\pm0.55$ | | | | |
| Random | 87.63 $\pm0.76$ | 93.43 $\pm0.65$ | 95.68 $\pm0.45$ | 97.30 $\pm0.13$ | 98.14 $\pm0.13$ | 58.74 $\pm0.76$ | 73.10 $\pm1.84$ | 80.95 $\pm0.66$ | 85.77 $\pm1.14$ | 87.64 $\pm0.72$ |
| Forgetting [48] | 15.58 $\pm0.47$ | 38.53 $\pm2.78$ | 75.85 $\pm1.69$ | 97.22 $\pm0.38$ | 98.11 $\pm0.04$ | 4.33 $\pm0.22$ | 22.33 $\pm0.31$ | 33.15 $\pm0.60$ | 64.43 $\pm1.23$ | 81.28 $\pm2.31$ |
| Entropy [11] | 41.46 $\pm3.46$ | 55.37 $\pm1.4$ | 69.04 $\pm1.16$ | 77.07 $\pm1.29$ | 91.98 $\pm0.83$ | 27.93 $\pm2.08$ | 41.69 $\pm0.73$ | 59.86 $\pm1.84$ | 78.69 $\pm2.13$ | 86.20 $\pm0.54$ |
| EL2N [39] | 14.16 $\pm1.14$ | 40.68 $\pm3.36$ | 81.25 $\pm3.22$ | 97.25 $\pm0.24$ | 98.16 $\pm0.30$ | 17.63 $\pm1.59$ | 23.24 $\pm1.88$ | 28.24 $\pm1.44$ | 37.58 $\pm1.53$ | 60.06 $\pm2.14$ |
| AUM [41] | 12.81 $\pm2.62$ | 35.10 $\pm3.46$ | 68.44 $\pm0.95$ | 93.76 $\pm1.89$ | 98.12 $\pm0.14$ | 4.56 $\pm0.18$ | 7.01 $\pm1.24$ | 22.13 $\pm1.86$ | 39.87 $\pm2.19$ | 65.93 $\pm1.61$ |
| CCS [61] | 88.05 $\pm0.62$ | 93.51 $\pm0.10$ | 95.58 $\pm0.32$ | 96.86 $\pm0.25$ | 97.18 $\pm0.08$ | 58.43 $\pm0.25$ | 71.73 $\pm0.83$ | 78.46 $\pm0.18$ | 83.64 $\pm0.55$ | 84.94 $\pm0.22$ |
| EVA (Ours) | 88.83 $\pm0.88$ | 94.43 $\pm1.32$ | 97.20 $\pm0.34$ | 98.27 $\pm0.57$ | 98.63 $\pm0.34$ | 61.23 $\pm0.75$ | 78.71 $\pm0.93$ | 83.11 $\pm0.72$ | 86.38 $\pm1.02$ | 88.77 $\pm0.43$ |

(left and right), heart, kidney (left and right), liver, lung (left and right), pancreas, and spleen. OrganAMNIST, previously known as OrganMNIST-Axial in MedMNIST v1 [53], consists of 58,830 axial view slices of abdominal CT images, distributed into 34,561 training, 6,491 validation, and 17,778 testing images. OrganSMNIST, formerly OrganMNIST-Sagittal, includes 25,211 abdominal CT images split into 13,932 training, 2,452 validation, and 8,827 testing images.

**Baselines and Networks.** We compare our method against six representative baselines, the latter five of which are state-of-the-art (SOTA) methods: 1) **Random**; 2) **Forgetting score** [48]; 3) **Entropy** [11]; 4) **EL2N** [39]; 5) **Area under the margin AUM)** [41]; 6) **Coverage-Centric Coreset Selection (CCS)** [61]. Details of these baselines are provided in the Supplementary material due to space limitations. The effectiveness of these strategies is evaluated based on their ability to select representative samples for coreset construction using various criteria. For all baselines except CCS, coresets are formed by pruning less important examples according to the respective importance metric.

The effectiveness of our method is primarily evaluated using ResNet-18 [18]. We also conduct cross-architecture generalization experiments with ResNet-50 [18], MobileNet [42] and VGGNet [45] to validate its robustness across different models. Further details are available in the Supplementary material.

**Implementation details.** To ensure fairness in our comparisons, we adhere to the experimental setup outlined in [61]. Our method is implemented using PyTorch [38] and all models are trained on an NVIDIA 3090 GPU. Unless specified otherwise, we utilize the same network architecture, ResNet-18, for both the coreset and the surrogate network on the full dataset. We maintain consistency in all hyperparameters and experimental settings before and after coreset selection. The surrogate network is trained for 200 epochs across all datasets. Initially, we train a network on the complete dataset to establish baseline performance. Subsequently,

we calculate the importance scores by assessing the variance of each sample's error vector across multiple epochs within a dual-window. As to the start epoch and end epoch of each window, we employ a grid search with a 10-step size ($K = 10$). This process helps us identify the most effective window combinations, denote as $(t_e, t_E)+(t_l, t_L)$ for different datasets and selecting rate $\alpha$.

## 4.2 Benchmark Evaluation Results

Our systematic comparison of EVA against other baselines, as detailed in Sec. 4.1, reveals its superior performance on the OrganSMNIST and OrganAMNIST medical datasets, particularly at more challenging selection rates. As shown in Tab. 1, our Evolution-aware VArianceapproach consistently achieves top-ranking performance, underscoring its robustness in coreset selection. In addition, on the OrganAMNIST dataset, EVA nearly matches the full dataset's performance at a 20% selection rate and surpasses it at 30%, highlighting its efficiency in utilizing smaller datasets. Notably, at extremely low selection rate of 2% and 5% on the OrganSMNIST dataset, EVA surpasses the Random baseline by a margin of 2.49% and 5.61%, respectively, illustrating its effectiveness even with severely limited data, establishing the method's capability to discern and retain the most influential samples for model training.

The baselines, including well-established SOTA methods, do not exhibit the same level of performance at these lower selection rates, often failing to exceed the benchmark set by random selection. This trend highlights the limitations of traditional coreset selection methods when dealing with the complexities of medical datasets.

Here, our experiments focus on low selection rates scenarios, but EVA also maintains competitive performance at high selection rates. Moreover, our methodology's effectiveness is not confined to medical imaging datasets alone. Preliminary experiments on widely recognized natural image datasets, such as CIFAR, corroborate that EVA stands out by surpassing most SOTA methods. Detailed

results from these additional experiments are documented in the Supplementary materials due to space constraints.

## 4.3 Ablation Study and Analysis

We delve into ablation studies to dissect the contributions of the variance and dual-window components in our method. By systematically removing each component and evaluating their impact on performance, we elucidate their individual roles in enhancing coreset selection accuracy. In this context, we partition our experiments into four conditions: Var-S, Exp-S, Var-D, and Exp-D. Here, Var-S denotes calculating variance in a single window, Exp-S represents computing expectation in a single window; Var-D indicates variance calculation in dual-window, and Exp-D signifies expectation computation in dual-window.

**Effectiveness of Variance.** In this section, to demonstrate the effectiveness of variance measurement, we display the test accuracy results of calculating the expectation and variance of the samples' error vectors within a single window or dual windows on different datasets. As shown in Fig. 3, these results were obtained under varied selection rates from 2% to 30%.

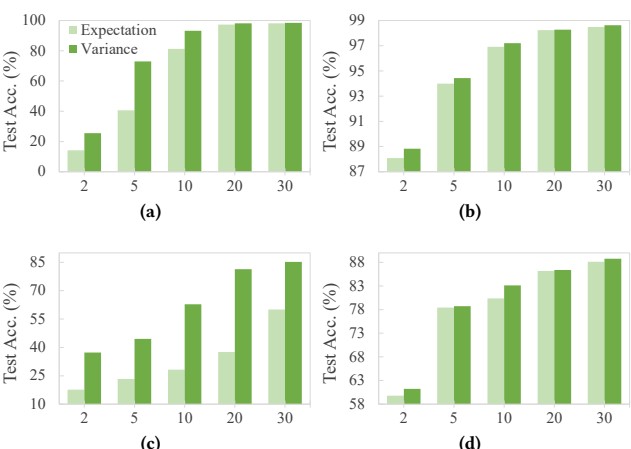

**Figure 3: Ablation study on the summary statistics. We validated the effectiveness of variance measurement under single-window and dual-window settings on OrganAMNIST (a)(b) and OrganSMNIST (c)(d). In (a) and (c), we contrast the Exp-S and Var-S strategies within an early 10-epoch window. (b) and (d) explore the Exp-D and Var-D strategies in dual-window setting.**

The first thing we notice is that, on both datasets, as the selection rate increases, the effectiveness of the models trained on the samples selected by both statistics tends to increase on the test set. This is intuitive because as the number of samples selected increases, the information richness of the selected samples are effectively preserved.

Besides, we can observe that at each selection rate, the variance measurement has better performance in coreset selection compared to the expectation measurement, and this advantage is especially significant at low selection rates. For example, in Fig. 3c, the test accuracy under Var-S is at least 20% higher than under Exp-S for

all compared selection rates. The consistent superiority of variance (Var-S and Var-D) suggests its robustness as a measure, further proving our previous points that (1) *Expectation* may mask variability within the data by averaging contributions, thereby potentially underrepresenting the underlying fluctuations. Samples with large expectation values may be consistently predicted incorrectly over many training iterations, indicating them too noisy/difficult and detrimental to the model's performance; (2) *Variance* captures the degree to which sample contributions fluctuate over training iterations. High variance in sample errors suggests that their influence on the model is not consistent but varies significantly, potentially due to their informative nature or because they are challenging for the model to learn. At low selection rates, samples with higher variance are indicative of a greater potential to contribute to the model's generalization ability, as they embody the critical challenges within the learning task.

**Effectiveness of dual-window.** In this section, we demonstrate the effectiveness of the dual window setting and analyze the results for different window combinations. First, we compare the performance of using single-window and dual-window on different datasets (as shown in Fig. 4). Similar to the former part, we utilized the variance and expectation of errors within single and dual windows as importance metrics. The results consistently demonstrate the advantage of dual windows over single window across all selection rates. This advantage, akin to the findings from the variance ablation experiments, is more pronounced at lower selection rates. For instance, on dataset OrganSMNIST, at selection rates of 2%, the variance calculated within dual windows exhibited an improvement of 2.29%, compared to the single-window approach, suggesting that employing dual-window calculation for scores enables more effective capturing of the diversity and variability of sample importance.

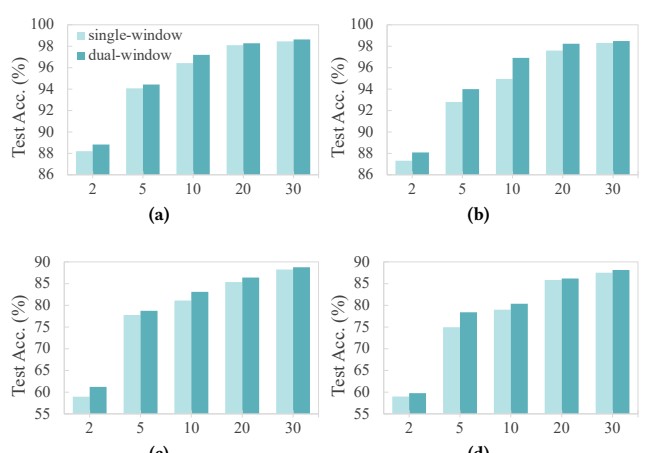

**Figure 4: Ablation study on the window setting. The results are obtained on OrganAMNIST (top row) and OrganSMNIST (bottom row). Performance of the Var-S versus Var-D strategies is illustrated in (a) and (c), while (b) and (d) show comparisons between Exp-S and Exp-D strategies.**

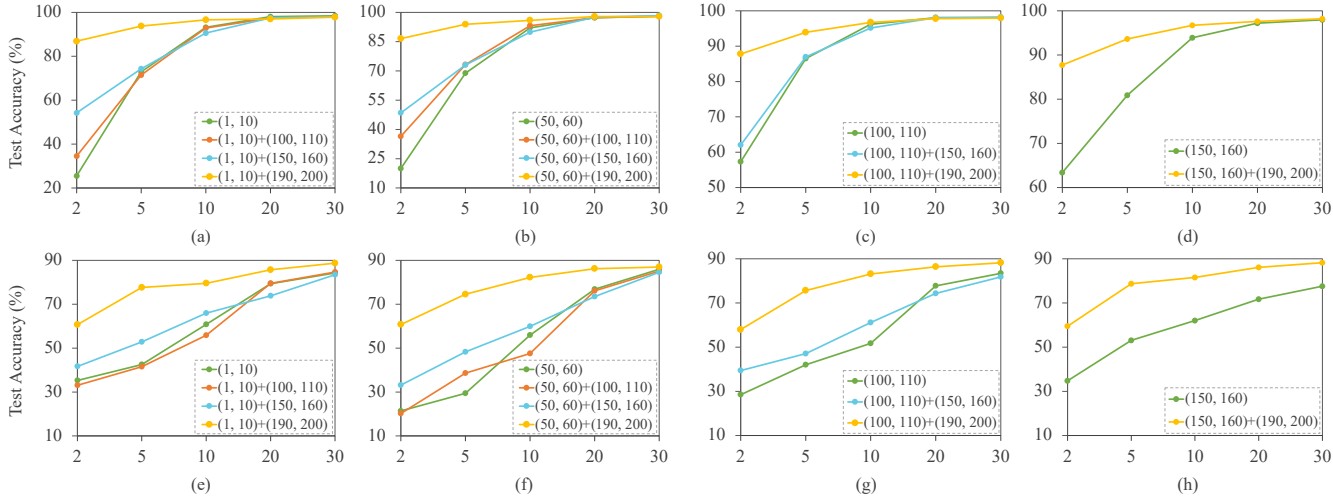

**Figure 5: Comparison of different window combinations. These windows represent different training phases. (a)-(d) show experimental results for OrganSMNIST, and (e)-(h) for OrganAMNIST, with each line depicting a unique window combination (single or dual windows).**

Moreover, in the dual-window setting, we further explore the effect of the combination of windows at different periods on model performance. Fig. 5 reveals two critical insights: (1) At a high compression rate of 2%, dual-window combinations show a definitive advantage over single-window ones on both datasets. This can be attributed to the dual-window's ability to encapsulate more diverse information from different stages of the training process, providing a broader perspective for coreset selection. (2) As the selection rate increases, allowing for larger data budgets, corresponding to the need of capturing a wider range of training dynamics. The implication here is that the windows selected for the dual-window setting should ideally come from a later stage in the training process, when the model has begun to stabilize and the samples are more reflective of the generalization capabilities required for the test. The results on OrganAMNIST suggests that the early dual-window stage may not be sufficient for selecting a more representative coreset.

## 5 RELATED WORKS

### 5.1 Medical Imaging

**Challenges in Medical Imaging with Deep Learning.**
Medical imaging technology has brought transformative advancements to the diagnosis of a variety of diseases in the past few decades, enabling earlier detection and the development of more personalized treatment plans. Deep learning (DL), in particular, has been widely used in various medical imaging tasks and has achieved remarkable success in many medical imaging applications [8, 36, 43, 62, 63], enhancing the accuracy of diagnoses through the innovative use of historical data [29].

Despite the substantial progress, integrating deep learning into medical imaging is fraught with challenges [22]. The effectiveness of DL in this context is largely dependent on the availability of large, well-annotated datasets tailored for specific tasks and reliant on advances in high-performance computing. The necessity for vast complex datasets introduces complications such as inconsistencies

in data quality, arising from variations in imaging equipment and protocols. Moreover, the extensive volume of medical data demands significant computational resources, posing logistical challenges for efficient processing [62]. Additionally, the inherent heterogeneity of medical images, characterized by a multimodal probability distribution, complicates the model training process by requiring algorithms capable of handling diverse visual features and patterns within the data. Another issue is the inter-class similarity and intra-class variation, as depicted in Fig. 6, where different diseases may appear similar, and the same disease may present differently across patients.

**MedMNIST: A Standardized Dataset for Biomedical Imaging.** To address some of these challenges, MedMNIST, a large-scale MNIST-like collection of standardized biomedical images, provides a comprehensive dataset for research and application. This dataset includes 12 datasets for 2D imaging and 6 for 3D, all pre-processed into 28x28 or 28x28x28 pixels with corresponding classification labels. MedMNIST encompasses primary data modalities in biomedical imaging, including abdominal CT, chest X-ray, breast ultrasound, and blood cell microscopy, making it an ideal choice for multi-modal machine learning in medical image analysis. Additionally, it supports various classification tasks such as binary/multi-class, ordinal regression, and multi-label classification, further establishing its utility for developing and testing deep learning models in medical imaging.

### 5.2 Dataset Compression

The proliferation of large-scale datasets in deep learning necessitates the compression of data size to meet specific requirements, such as computational efficiency and storage constraints. Therefore, the identification of key samples serves a fundamental role not only in dataset pruning but also across a spectrum of machine learning tasks, such as active learning [1, 4, 14], where the model is trained iteratively on a subset of the dataset, and only the most informative

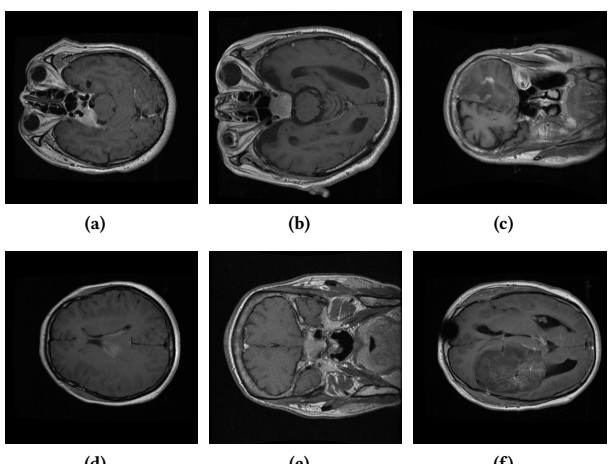

**(a)**    **(b)**    **(c)**

**(d)**    **(e)**    **(f)**

**Figure 6: Examples of the intra-class variation and inter-class similarity in medical image classification. These axial brain tumor images come from the public dataset provided by Jun Cheng et al.[9]. Each column respectively represents a brain tumor category: meningioma (a)(d), pituitary (b)(e), and glioma (c)(f). The variation within the same category can be noticed by observing the two instances in each column. Furthermore, the similarity between different classes is illustrated by comparing (a)(b), (c)(e), and (d)(f).**

samples are selected for inclusion in subsequent training rounds. Techniques such as uncertainty sampling and query-by-committee have been proposed to select data samples that are most beneficial for model improvement. Continual learning [57], where a memory buffer is maintained to store informative training samples from previous tasks for rehearsal in future tasks. And other problems like noisy learning [35], clustering [2], semi-supervised learning [5], and unsupervised learning [10].

Dataset pruning, also known as coreset selection, can generally be categorized into several groups: Score-based techniques [11, 15, 34, 39, 47, 48], methods driven by coverage or diversity considerations [44, 50, 51], and strategies grounded in optimization [21, 23–25, 27, 35, 49, 55]. Specifically, score-based techniques first assign an importance score to each training sample based on its influence over a specific permanence metric during model training. The samples are then sorted by their scores, and those within a certain range are selected to construct the coreset.

Besides, in the sphere of data-efficient deep learning, associated topics include techniques like data distillation [6, 30] and data condensation [12, 26, 30], which seeks to condense the knowledge contained in a large dataset into a smaller, distilled dataset. This technique often involves training a smaller "student" model to mimic the behavior of a larger "teacher" model, effectively transferring the knowledge from the larger dataset to the distilled one. Similarly, most distillation methods are evaluated on natural image datasets and their effectiveness lack comprehensive verification on medical datasets. To the best of our knowledge, a recent work [59] propose a comprehensive benchmark to evaluate the medical image dataset distillation.

## 6 LIMITATION & FUTURE WORK

While our EVA coreset selection strategy demonstrates superior performance in high compression scenarios, as evidenced by the comparative analysis presented in Tab. 1, it's important to acknowledge the limitations that the level of accuracy achieved in scenarios demanding extreme compression may not fully meet the rigorous standards necessary for medical diagnostics. Medical imaging tasks often require the highest degree of precision due to their direct impact on patient care, and there remains room for improvement in ensuring that the selected coresets are not only statistically representative but also clinically relevant.

Additionally, our current approach does not incorporate data from different modalities, which is essential in smart healthcare diagnostic systems. Such systems typically combine various types of data, including medical images, electronic health records (EHRs), patient interview descriptions, and pathology reports, for holistic analysis to enhance diagnostic accuracy.

Future research could focus on exploring different compression limits for various datasets to find the optimal balance between accuracy and efficiency. This would involve systematically determining how much data can be pruned while still maintaining sufficient performance levels for clinical applications. Moreover, there is a promising avenue to extend our work by integrating multimodal data, which would align well with the ongoing trends in applying large language models (LLMs) and other advanced AI techniques in healthcare. Such integration could enhance the robustness and applicability of our coreset selection strategy, particularly in systems where diverse types of data need to be synthesized for effective decision-making.

## 7 CONCLUSION

In this paper, we identify the limitations of existing coreset selection methods in capturing the evolutionary nature of model training and the fluctuations in sample importance within medical image datasets. To address this challenge, we introduced a novel sample scoring strategy, Evolution-aware VAriance (EVA), which incorporates a dual-window method to consider the training dynamics at different stages and employs a variance measurement of samples' error vectors for a more precise evaluation of sample importance. Extensive evaluations on various datasets and networks demonstrate the superior performance of our proposed EVA strategy.

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
