# OpenReview forum: "Evolution-aware VAriance (EVA) Coreset Selection for Medical Image Classification"
_acmmm.org/ACMMM/2024/Conference — MM2024 Oral_

### Official Review · Reviewer_7uVi · 2024-05-22

**Rating:** 4
**Confidence:** 2

**Summary:**

*Problem context*: Medical datasets are large and resource intensive to train models for (data transfer, processing, etc).

*Gap(s) in literature*: Coreset selection is under explored in medical imaging contexts. Current methods do not capture the variability of sample contributions throughout the training process. SOTA methods often perform worse than random selection of data on medical imaging datasets.

*Proposed solution*:
“EVA reflects the epoch-level fluctuation by calculating the variance of error-based scores”

All samples have their error captured throughout two time periods during training: early phase and late phase. Purportedly to capture the varying importance of certain samples based on the features they contain. The variance is calculated for each sample across the two windows. The most variable samples are considered the most important for learning, and so are selected as part of the coreset.

*Contributions*:
Novel theory on sample importance.
New method for selecting a subset of training samples based on error-variance throughout training.

**Strengths:**

EVA outperforms SOTA methods on both natural and medical imaging datasets, in some cases by significant margins.

Experiments are well described and support the aims of the work.

Variance based selection has a much more significant impact on performance at much lower selection rates than expectation-based methods.

**Limitations:**

How does the dual-window variance compare to calculating the variance across the entire training process? While this would be resource intensive, it might be worth detailing to show that the dual-window approach accurately reflects the variance across the entire training process.

Structure: related works should be placed in the introduction not at the end of the article.

How useful is this type of methodology in practise? I may be missing something here, but to be able to use this method to select the data to train a model with, one must be able to train models on the dataset to begin with, which appears to defeat the purpose. Unless the ultimate goal is to send the subset of data around for other models to be trained later. The possible improvement in performance with the subset selection though is a strong argument, however.

**Suitability:**

2

---

### Official Review · Reviewer_MwjM · 2024-05-23

**Rating:** 6
**Confidence:** 4

**Summary:**

In this paper, the authors propose a new coreset selection method for the medical field called Evolution-aware Variance (EVA). EVA enhances the selection of important data for classification by considering both intra-class and inter-class variance. Using a dual-window approach, it captures the model's training evolution and measures sample importance more precisely. Experiments on medical image datasets show EVA's superiority, especially with limited training images. EVA achieves 98.27% accuracy with only 10% of the training data, compared to 97.20% with the full dataset. Additionally, EVA outperforms random selection by 5.61% at a 5% selection rate, highlighting its effectiveness for efficient medical image analysis.

**Strengths:**

- The paper addresses the critical issue of dataset transfer costs in the field of medical imaging.
- The proposed method has a simple structure yet appears to function effectively.
- The reliability of the method is supported by diverse experiments, including an ablation study, demonstrating its robustness from multiple perspectives.

**Limitations:**

- I am interested in the potential for overfitting when using small training datasets. It seems necessary to carefully select the training set to avoid overfitting. How should this be handled within this framework?
- Which aspects of the method are specifically tailored to the nature of medical images? The proposed method appears to be a highly versatile approach that could also be effective for general images.

**Suitability:**

3

---

### Official Review · Reviewer_BSUQ · 2024-05-31

**Rating:** 5
**Confidence:** 4

**Summary:**

This paper addresses the challenge of managing high-dimensional medical imaging data and performing reliable analysis in resource-limited environments. The authors propose a novel coreset selection strategy called Evolution-aware VAriance (EVA), which captures the evolutionary process of model training using a dual-window approach and measures sample importance through variance. The paper demonstrates that EVA outperforms state-of-the-art methods, achieving high accuracy with significantly reduced training data. For instance, EVA achieves 98.27% accuracy with only 10% of the training data, compared to 97.20% with the full training set.

**Strengths:**

This paper introduces a novel coreset selection strategy specifically designed for the complexities of medical image datasets. The Evolution-aware VAriance (EVA) approach is innovative in that it captures the dynamic changes in sample importance throughout the training process, addressing the shortcomings of traditional snapshot-based methods. This dual-window approach provides a more accurate reflection of sample importance by considering different training stages and the variance in error vectors.

The methodology is technically sound, employing a well-justified dual-window approach to track sample importance across different training stages. The use of variance measurement to capture fluctuations in sample importance is a robust and theoretically solid approach, enhancing the precision of the evaluation.

The paper includes extensive experimental validation on the OrganAMNIST and OrganSMNIST datasets. The results consistently demonstrate the superiority of the EVA method over several state-of-the-art coreset selection techniques. The authors provide detailed performance comparisons, highlighting EVA's ability to maintain high accuracy even at significantly reduced data volumes. The method's effectiveness is further demonstrated through various selection rates, underscoring its robustness and adaptability.

The paper is well-written and clearly structured, making the complex methodology accessible to a broad audience.

**Limitations:**

The dual-window approach used in EVA focuses on capturing training dynamics at early and late stages of model training. This strategy might bias the sample selection towards these specific phases, potentially neglecting samples that gain importance in intermediate stages of training. This could result in a coreset that does not fully represent the diverse characteristics of the entire dataset.

Medical images often contain classes with high visual similarity, making it difficult to distinguish between different conditions. The paper does not extensively discuss how EVA handles this issue of inter-class similarity.

The datasets used in this study are of moderate size. The scalability of EVA to larger datasets, which are common in medical imaging, remains uncertain. Larger datasets may introduce additional challenges such as increased computational load and memory requirements.

 While EVA aims to reduce computational costs by selecting a smaller, more representative coreset, the dual-window and variance calculations add complexity. The paper lacks a detailed analysis of the computational overhead introduced by these additional steps.

**Suitability:**

2

---

### Meta-Review · Area_Chair_Qz3t · 2024-06-23

**Recommendation:** Accept (Oral)
**Confidence:** 5

**Metareview:**

The paper addresses a critical challenge in medical imaging: efficient handling and analysis of high-dimensional data in resource-limited environments. The proposed Evolution-aware VAriance (EVA) method demonstrates good methodological rigor and thorough experimental validation. The dual-window approach and variance-based selection method are explained with detailed theoretical justifications, and the experimental results comprehensively support the authors' claims.

EVA introduces a novel approach to coreset selection by capturing the evolutionary process of model training through a dual-window approach and reflecting the fluctuation of sample importance more precisely through variance measurement. This innovation addresses the limitations of existing coreset selection methods, particularly for medical image datasets, which often exhibit high intra-class variation and inter-class similarity.

The proposed method has significant implications for medical image analysis, especially in resource-constrained environments. EVA's ability to achieve high accuracy with significantly reduced training data can lead to more efficient storage, transmission, and computational costs. The extensive experimental validation on medical image datasets demonstrates EVA's potential to outperform state-of-the-art methods, highlighting its practical relevance and importance.

Pros:
+ The paper is well-organized and written.
+ The introduction of the dual-window approach and variance-based selection method is a novel contribution to the field of coreset selection, particularly tailored for the complexities of medical image datasets.
+ The paper includes detailed experiments on multiple medical image datasets, demonstrating the robustness and effectiveness of EVA.

Cons:
+ The scalability of EVA to larger datasets is not fully explored.
+ The practical utility of the method could be further clarified, particularly regarding the need to train models to select the subset of data initially.

Given the three highly positive reviews, I recommend accepting this paper for an Oral presentation.